# Psychometric Properties of the Identity Bubble Reinforcement Scale (IBRS) in a Sample of Chilean Adolescent Students

**DOI:** 10.3390/children12111545

**Published:** 2025-11-14

**Authors:** Karina Polanco-Levicán, José Luis Gálvez-Nieto, Sonia Salvo-Garrido, Ignacio Norambuena-Paredes, Nathaly Vera-Gajardo

**Affiliations:** 1Facultad de Educación, Universidad Autónoma de Chile, Temuco 4810101, Chile; karina.polanco@cloud.uautonoma.cl (K.P.-L.); nathaly.vera@cloud.uautonoma.cl (N.V.-G.); 2Departamento de Trabajo Social, Universidad de La Frontera, Temuco 4780000, Chile; ignacio.norambuena@ufrontera.cl; 3Departamento de Matemática y Estadística, Universidad de La Frontera, Temuco 4780000, Chile; sonia.salvo@ufrontera.cl

**Keywords:** identity bubble, social networks, validity, reliability, adolescence

## Abstract

**Highlights:**

**What are the main findings?**

**What is the implication of the main finding?**

**Abstract:**

Background/Aim: Social networks have transformed the traditional dynamics of identity construction in adolescence, allowing users to select content and interact with others who share similar views, thereby reinforcing a sense of belonging to homogeneous groups. Given the growing influence of digital interaction on social identity among youth, psychometrically sound instruments are needed to measure this process. This study aimed to evaluate the psychometric properties of both the 9-item (IBRS-9) and 6-item (IBRS-6) versions of the Identity Bubble Reinforcement Scale in a large sample of Chilean adolescent students. Methods: A cross-sectional design was used with 4096 participants (50.8% male, 47.8% female, 1.4% other; M = 15.82, SD = 1.30) from 41 secondary schools across Chile. Confirmatory factor analyses (CFAs) tested factorial validity, and internal consistency and external criterion validity were examined. Measurement invariance was assessed across sex, social media use, internet use, and age. Analyses were conducted using the WLSMV (Weighted Least Squares Mean and Variance Adjusted), and model evaluation was based on conventional goodness-of-fit indices. Results: CFAs supported the factorial validity of both IBRS versions, showing reliability and external criterion validity. Model fit indices indicated good fit for both scales. Invariance analyses confirmed factorial stability up to the strict level across all subgroups, indicating consistent psychometric performance. Conclusions: The IBRS-9 and IBRS-6 are valid and reliable instruments for assessing identity bubble reinforcement among Chilean adolescents, providing evidence of factorial stability and applicability for research and educational and psychosocial interventions. Their validated structure provides a consistent basis for examining social identity processes related to digital interaction.

## 1. Introduction

Adolescents have been born and raised in a time in which social networks are present and, therefore, have observed how their environment communicates not only face-to-face, but also how online interactions are part of the usual communication of a large portion of people [1,2]. Young people may even prefer this type of communication compared to in-person interactions to relate to others and form social groups [3]. In this way, adolescents socialise in an interconnected world where boundaries are blurred and it is possible to interact at any time [4]. This type of interaction, as well as participation in digital platforms, influences psychosocial processes in various ways, considering the adolescents’ personal characteristics [5,6]. In this sense, social networks also change the way social groups are formed, affecting the construction of social identity [7,8,9,10].

Social networks are digital platforms that allow massive interaction in verbal and visual forms through the internet [11]. In the case of adolescents, they are both content creators and consumers [12,13,14]. Adolescents use social networks for various purposes, including entertainment and fun, obtaining information, meeting new people, and even avoiding and escaping feelings and situations that evoke negative emotions [15,16]. In this sense, the virtual context is an essential part of adolescents’ lives, and social networks have progressively been integrated into the ways they communicate their interests, playing a significant role in identity development [17]. Furthermore, social networks enable the creation of bonds between people, allowing young people to initiate and maintain affective relationships of varying levels of closeness and relevance, thereby encouraging their participation [18].

Thus, social media enables the generation of social connections and can foster a sense of belonging among adolescents [19]. Therefore, they constitute a context for the construction of adolescents’ social identity, allowing for greater exploration considering the characteristics of this medium and its possibilities [17,20,21]. In this regard, technology and creativity offer new opportunities for expression and learning, contributing to young people’s perceived well-being across various domains, as evidenced during the COVID-19 pandemic. Creativity is essential in adaptation processes and plays a significant role in adolescent development and throughout the lifespan, favouring the ability to confront complex and unpredictable situations. Creative thinking fosters the formation of flexible identities, which is crucial in the face of changing contexts. Consequently, the educational system, by favouring these elements, supports adolescent development and their adaptation to digital contexts [22].

Social identity is understood as the knowledge and emotional appraisal that a person perceives regarding their participation in one or more groups [23]. Young people define their identity considering different experiences in various contexts, with those groups that are more easily accessible exerting greater influence on the construction of their identity [24,25,26]. The above is relevant considering that social media is frequently used, even though at this stage of development, there may be greater parental regulation compared to later stages [27,28].

It is significant that during adolescence, psychosocial changes occur that are associated with the need for autonomy, belonging, and acceptance by peers, which is influenced by the digital context [21]. Peer groups may pressure adolescents to be available online, influencing the perception of friendship quality [29], fostering a more active use of social media [30]. In the case of those who show greater dependence on feedback from others on social media, they may experience higher levels of depression [31]. In this sense, the use of social media has shown both positive and negative influences at the cognitive and affective levels [32], entailing risks for adolescents [33]. Consequently, this context is fundamental for understanding young people’s interactions today [29].

According to Aran-Ramspott et al. [34], gender differences are evident in the way social media content is experienced and given meaning, which is closely linked to cultural aspects. In particular, women tend to perceive greater pressure related to physical appearance and clothing, reflecting a significant influence of platforms such as Instagram and TikTok on the construction of self-image and self-esteem. In contrast, men report a different experience, associating exposure on social media with feelings of self-affirmation, fun, and social prestige. Likewise, it is noted that adolescent girls spend more time on their smartphones and social media than males, who may prefer other types of interactive digital media [35].

Additionally, gender differences associated with the symptomatology of problematic internet use are reported. Specifically, women present greater social withdrawal compared to men, who show greater interpersonal conflicts linked to problematic internet use [36]. Other differences linked to social media use and age evidence a significant inverse relationship observed between estimated social media use and life satisfaction indices one year later. This phenomenon occurs during periods of heightened sensitivity when interacting with these types of digital platforms, which vary by age for men (14–15 and 19 years) and women (11–13 and 19 years) [37]. Therefore, it is relevant to consider adolescents’ ages, as identity formation develops progressively. Identity evolves gradually throughout adolescence, simultaneously showing stability and systematic maturation in the processes of exploration and commitment in the construction of the self [38].

The Identity Bubble Model [7,37] recognises that virtual spaces have become a meeting point for adolescents, fostering the formation of social communities that influence identity construction [19,37,38]. The contributions of Social Identity Theory [38] form the basis for the Identity Bubble Model [37], which provides a new understanding of the formation of social identity by considering the virtual context, the way people interact, and the information they share with their group [37].

The Identity Bubble Model [7,37] proposes three factors that influence the construction of social identity, which are related to how social media functions, how interactions occur, and the type of information that tends to be more accessible. The factors are the following: social identification, which is linked to the commitment and sense of belonging experienced by young people toward groups on social media. On the other hand, homophily relates to characteristics at the personal, social, and economic levels, among others, shared by the people who make up the groups. This limits the possibility of interacting with different people, which is essential for reducing prejudice toward those who are different and for improving coexistence [39,40]. Another important element is confirmation bias, which is promoted by content recommendation algorithms, by social groups similar to the user, and by the ability to select the people with whom one wishes to share, leading them to be exposed to information similar to their own beliefs [7,37].

Considering the above, the Identity Bubble Reinforcement Scale [7] is presented, which includes three correlated dimensions: social identification, homophily, and confirmation bias. The authors [7] present two versions of the scale, differing in the number of items: one with six items (IBRS-6) and one with nine items (IBRS-9). Both show adequate psychometric indicators in samples of young people from Finland, the United States, Spain, and South Korea. In addition, evidence of measurement invariance is presented between different countries for the IBRS-6 scale, reaching the level of metric invariance. In Chile, evidence of measurement invariance has been reported using the IBRS-9 in a population of university students. Specifically, it is worth noting that the three-dimensional structure proposed by the authors [7] is confirmed, as evidenced by adequate validity indices and reliability levels. Factorial invariance was tested considering sex (male/female) and social media use (low use/high use). In both cases, scalar invariance was achieved, leading to the conclusion that intercepts are equivalent across sex and social media use [41].

In addition, online disinhibition constitutes another key process for understanding behaviour in virtual environments. The social identity of group members and online disinhibition are processes that interact with the external environment, in this case, the online context, where anonymity fosters behaviours linked to online aggression. When group members feel committed to and value this belonging, they may display behaviours that they would not adopt individually [42]. It is also observed that homophily, as part of social identity, significantly influences users’ negative behaviour on the internet [43]. In this sense, refs. [43,44,45] note that anonymity, invisibility, and the perception of safety on the internet reduce usual social restraints, which increases the expression of freer or more transgressive behaviours. The Measure of Online Disinhibition (MOD) has demonstrated adequate levels of validity and reliability in Chilean populations [46], and it has been linked to increased online participation and expression [47]. Together, the IBRS-9 and the MOD provide a complementary framework for analysing how adolescents construct and express their identity in digital spaces.

Along the same lines, virtual context characteristics influence internet trolling behaviour, which is defined as posting offensive and provocative comments with the intent to generate discussion, cause harm, or provide entertainment [48]. This antisocial behaviour affects adolescents [49] and might be linked to seeking recreation and social approval [50]. Moreover, users experiencing social media fatigue may interpret others’ behaviours as hostile, increasing the probability of engaging in trolling [51]. The foregoing is based on the understanding that the number of hours spent on the internet increases participation in online trolling [52]. It should be noted that social identity and the group belonging processes emerging in anonymous digital contexts foster behavioural disinhibition and, consequently, participation in trolling [53].

Furthermore, consideration is given not only to the duration of time spent on social media, but also to the subjective experience associated with appearance-related concerns. In this respect, recent constructs have emerged that extend traditional models of self-objectification within the digital context, such as Appearance-Related Social Media Consciousness (ASMC) [54,55]. This construct incorporates the specific psychological dynamics of social media, in which individuals maintain a constant awareness of how an online audience may evaluate their appearance. The Appearance-Related Social Media Consciousness Scale (ASMCS) operationalises this construct and assesses the degree to which thoughts, emotions, and behaviours are influenced by this perceived social evaluation. The Italian version (ASMCS-I) [56] shows satisfactory psychometric properties, including reliability and validity. This approach offers a complementary framework for understanding the dynamics of self-awareness and digital self-presentation that may coexist with the reinforcement processes of identity bubbles.

Concerning the scale’s applications, it can be noted that the IBRS-6 was utilised in a sample of Finnish adolescents and young adults, evidencing that social homophily is associated with greater online offending hatred among those presenting internalising symptoms. Also, perpetrators tend to follow group norms when a shared identity is activated, reinforcing identity bubbles [7]. Additionally, this scale was applied in a sample of young people aged 15 to 25 from different countries (the United States, South Korea, Spain, and Finland), observing that participation in online gambling communities was more strongly and significantly associated with problematic gambling [57]. On the other hand, a study with Chinese medical staff found that reinforcing the identity bubble on social media is positively associated with general self-efficacy and the happiness index, with this relationship mediated by self-efficacy [58].

Regarding the applications of the IBRS-9, it is worth noting that it has been utilised in various studies [59,60,61,62]. The study [62] reports that greater social identification is associated with higher levels of cyberaggression in adolescents and emerging adults (aged 15 to 25 years). This sample included countries such as Finland, Spain, South Korea, and the United States. The study [50] in a longitudinal and transnational study that included six countries, concluded that identity bubbles are associated with addictive internet use. This instrument was also used in a study linking identity bubbles with convergent self-esteem on social media among adolescents [8], both of which are positively related, considering that this type of self-esteem is conditioned by feedback from other users (likes, comments, among others). On the other hand, a relationship has been observed between online cliques and exposure to gambling-related content, since identification with these groups, along with similar information shared and reinforced by recommendation algorithms, fosters certain behaviours related to the topic [9]. In this sense, online gambling problems increase when users are within identity bubbles. These types of bubbles are relevant in processes where non-problematic leisure activities turn into addictive behaviours [10,62].

The relevance of adapting and validating this instrument in adolescent populations is related to the developmental stage they are in, which is characterised by the need to establish bonds of friendship with their peers, and the fact that the virtual context has become a space where they participate actively. In Chile, the use of the internet and social media has progressively expanded, with 92% of students having access to the internet at schools, and 79% from their homes, in addition to obtaining their first cellphone at an average age of 8.9 years [60]. Regarding social media, 25% of students aged 9 to 10, 32% aged 11 to 13, and 41% of adolescents aged 14 to 17 report “Participating in a virtual community where there are people who share their interests or hobbies.” This is despite the recommended age for social media use being 13 years for platforms such as Instagram, Facebook, and YouTube [63,64]. Thus, adolescents are exposed to different content and are part of groups that generate and share information that influences the behaviour, emotions, and beliefs of their members [64,65,66].

Therefore, further research is required to investigate the influence of social media on the construction of social identity, considering the commitment and sense of belonging generated toward virtual groups [21]. This, considering that it is beneficial for identities to be flexible in order to adapt to diverse groups and individuals, while identifying with limited groups may foster discrimination and, more generally, social prejudice [24,25]. For this reason, it is relevant to adapt and validate the Identity Bubble Scale for Chilean adolescents, as this research will provide evidence of validity and reliability in this population. This scale will enable the evaluation of social identity construction linked to participation in online groups, taking into account the relevance of the virtual context and its characteristics for today’s youth. This builds upon previous contributions, in which the psychometric properties of the Identity Bubble Scale were evaluated among Chilean university students [42].

Consequently, the following hypotheses were proposed: the scores of the Identity Bubble Reinforcement Scale will show a three-factor structure with correlated dimensions—Social Identification, Homophily, and Confirmation Bias—along with adequate levels of reliability in the Chilean context; the scores of the scale will remain invariant up to the scalar invariance level according to the variables sex, social media use, and internet use. Finally, the aim of this study was to evaluate the psychometric properties of both the 9-item (IBRS-9) and 6-item (IBRS-6) versions of the Identity Bubble Reinforcement Scale in a large sample of Chilean adolescent students.

## 2. Method

### 2.1. Participants

The population studied consisted of 322,043 students from public, subsidised private, and private schools, representing five macro-zones of Chile. Participants were selected using a probabilistic, stratified sample with a reliability of 99.7%, a margin of error of 2.6%, and a variance of *p* = q = 0.5. Table 1 shows the composition of the sample, which consisted of 4096 students of both sexes (50.8% men, 47.8% women, and 1.4% other), with a mean age of 15.82 years (Sd = 1.30). These students came from 41 high schools in Chile.

### 2.2. Instruments

First, a sociodemographic questionnaire was administered, which inquired about background information, including age, gender, internet use, social media use, and other relevant variables.

Additionally, the adapted version for Chilean university students of the Identity Bubble Reinforcement Scale IBRS [40] was applied. The IBRS is an instrument that measures the construction of social identity considering the characteristics of social media [7].

According to Polanco et al. [59], the adaptation process for this instrument was carried out in accordance with the guidelines of the International Test Commission [27]. The original authors of the scale [7] sent the Spanish version for its adaptation in Chile [59]. Subsequently, an expert panel reviewed the items and concluded that no modifications were necessary, given that they provided adequate clarity and relevance to the Chilean social and cultural context. Finally, a qualitative pilot study was conducted with university students to confirm the instrument’s comprehension and suitability in this population.

The IBRS is a self-report instrument consisting of 9 items to be answered on a five-point ordinal response scale (1 = strongly disagree, 5 = strongly agree). The IBRS has a structure of three correlated factors named as follows:

Social identity (item 1, e.g., In social media, I belong to a community or communities that are important part of my identity, Homophily (item 3, e.g., In social media, I prefer interacting with people who are like me), Confirmation bias (item 8, e.g., In social media, I trust the information that is shared with me).

Subsequently, the adapted version for Chilean university students of the Measure of Online Disinhibition MOD [48] was applied. The MOD scale is an instrument that evaluates the user’s perception of the decrease in their own behavioural restrictions when online [45]. This instrument is a self-report scale consisting of 12 items, which are answered using a five-point ordinal scale (1 = not at all like me, 5 = very much like me). Psychometric studies have shown that the MOD has a one-factor structure (e.g., item 2, “I am more able to discuss controversial issues online than I am in person”). The MOD has a one-dimensional factor structure, and psychometric studies have shown adequate levels of reliability and validity [45].

Finally, the Global Assessment of Internet Trolling (GAIT) [48] was administered. The GAIT is a self-report measure designed to assess individuals’ tendency to engage in trolling behaviours across online platforms. This instrument consists of four items rated on a five-point Likert scale (1 = strongly disagree, 5 = strongly agree). Previous research indicates that the GAIT has a one-factor structure (e.g., item 2: “I like to troll people in forums or the comments section of websites”), with psychometric evidence demonstrating satisfactory reliability and validity [48]. The adaptation of the GAIT for Chilean adolescents is currently underway.

### 2.3. Procedures

To apply the instruments, we first contacted the directors of the participating educational institutions and obtained their signed agreement to access the sample. Parents or legal guardians were subsequently asked to provide informed consent, and once these authorizations were in place, students gave their informed assent to participate. The study protocols and ethical safeguards were reviewed and approved by the Ethics Committee of the Universidad de La Frontera, Chile.

### 2.4. Data Analysis

Descriptive statistics (mean, median, standard deviation, interquartile range, skewness, and kurtosis) were computed using SPSS version 25. Confirmatory factor analyses for both the IBRS-9 and IBRS-6 were conducted in Mplus version 8.1 using the weighted least squares mean and variance adjusted estimator (WLSMV) with TYPE = COMPLEX to account for clustered data [67]. Competing models (three correlated factors and second-order) were compared using the DIFFTEST procedure. Model fit was evaluated with the following criteria: Comparative Fit Index (CFI) and Tucker–Lewis Index (TLI) values equal to or greater than 0.90 (acceptable) and 0.95 (good), Root Mean Square Error of Approximation (RMSEA) values equal to or lower than 0.08 (acceptable) and 0.06 (good), and standardised Root Mean Square Residual (SRMR) values equal to or lower than 0.08 [68,69]. Measurement invariance across sex, social media use, internet use, and age was tested sequentially (configural, metric, scalar, and strict). Invariance was accepted when changes in CFI were equal to or lower than 0.010, changes in TLI were equal to or lower than 0.010, and changes in RMSEA were equal to or lower than 0.015. Convergent and discriminant validity were assessed through Composite Reliability (CR), Average Variance Extracted (AVE equal to or greater than 0.50), and the Heterotrait–Monotrait ratio (HTMT lower than 0.85 to 0.90) [70], as well as Pearson correlations between IBRS factors and theoretically related constructs (Online Disinhibition and Internet Trolling). Internal consistency reliability was estimated with McDonald’s omega and Cronbach’s alpha, including their 95 percent confidence intervals. Values equal to or greater than 0.70 were interpreted as acceptable for research use [70,71,72]. All statistical tests were evaluated at a significance level of p lower than 0.01.

## 3. Results

### 3.1. Descriptive Analysis

The descriptive statistics for the scale are shown in Table 2. The highest mean was obtained by item 9 “In social media, I can keep myself well informed” (M = 3.85, Md = 4.00, Sd = 1.04, IQR = 2.00), while the item with the lowest mean is item 7, “In social media, I belong to a community or communities that I can commit to” (M = 2.35, Md = 2.00, Sd = 1.21, IQR = 2.00).

### 3.2. Confirmatory Factor Analysis

To assess the dimensionality of the IBRS, a series of competing confirmatory factor models were tested using the WLSMV with a TYPE = COMPLEX correction to account for the cluster sampling design (students nested within educational institutions). Table 3 presents the fit indices of the three-correlated-factor and second-order models for both the 9-item and 6-item versions of the IBRS.

For the IBRS-9, the three-correlated-factor model showed an excellent fit to the data (WLSMV-χ^2^(24) = 429.746, *p* < 0.001; CFI = 0.978; TLI = 0.967; RMSEA = 0.061; SRMR = 0.044). A second-order model, with a general factor loading on the three specific factors, also demonstrated excellent fit (WLSMV-χ^2^(25) = 448.947, *p* < 0.001; CFI = 0.977; TLI = 0.967; RMSEA = 0.061; SRMR = 0.045). Whilst the DIFFTEST indicated a statistically significant difference between the models (Δχ^2^(1) = 34.960, *p* < 0.001), the change in fit indices was negligible (ΔCFI = −0.001; ΔRMSEA = 0.000). Therefore, the second-order model was accepted owing to its parsimony and equivalent fit.

Similarly, for the IBRS-6, both the three-correlated-factor model (WLSMV-χ^2^(6) = 31.521, *p* < 0.001; CFI = 0.998; TLI = 0.994; RMSEA = 0.031; SRMR = 0.013) and the second-order model (WLSMV-χ^2^(7) = 52.980, *p* < 0.001; CFI = 0.996; TLI = 0.991; RMSEA = 0.038; SRMR = 0.018) exhibited excellent fit. Whilst DIFFTEST indicated a significant difference (Δχ^2^(1) = 15.950, *p* = 0.0001), the change in CFI (ΔCFI = −0.002) and RMSEA (ΔRMSEA = 0.007) was negligible, again supporting the adoption of the second-order structure as the most parsimonious model.

Altogether, these findings support the existence of an underlying second-order factorial structure for both versions of the IBRS, with a general identity bubble reinforcement factor explaining the covariance among the three specific dimensions: social identity, homophily, and confirmation bias.

Within the framework of Confirmatory Factor Analysis (CFA), Construct Reliability (CR) and Average Variance Extracted (AVE) were calculated for each dimension of the IBRS-9 and IBRS-6, indicating adequate levels of convergent validity. Discriminant validity was assessed using the Heterotrait–Monotrait (HTMT) ratio. The results (Appendix A) confirmed adequate discriminant validity between the dimensions of both versions. In the second-order models, hierarchical omega (ωh) and total omega (ωt), coefficients were estimated, which indicated a significant contribution of the general factor relative to the specific dimensions (see Appendix A).

### 3.3. Factorial Invariance

To evaluate measurement model equivalence across the analysed groups, a sequential factorial invariance analysis was conducted using the weighted least squares mean and variance adjusted (WLSMV) robust estimator. Four grouping variables were considered: sex, social media use, internet use, and age (Table 4). In all cases, the configural model (M0) showed satisfactory fit indices, indicating that the factorial structure composed of items and latent dimensions was stable across the compared groups. When metric invariance (M1) constraints were applied, assuming equal factor loadings, the observed changes in fit indices were minimal and within the recommended cut-off values. This result suggests that the relationships between the observed items and their corresponding latent factors are equivalent across groups. Upon assessing scalar invariance (M2) by constraining the factor loadings and item thresholds, the fit indices remained stable, supporting the comparability of latent means across the groups defined by sex, social media use, internet use, and age. Finally, the strict invariance model (M3), which also constrained the residual variances, maintained an adequate fit, confirming the equivalence of measurement errors across groups. Altogether, these results provide evidence of configural, metric, scalar, and strict invariance for the IBRS-9 and IBRS-6 models across all grouping variables.

### 3.4. Convergent Validity

To evaluate the convergent validity of the Identity Bubble Reinforcement Scale (IBRS), bivariate Pearson correlations were calculated between its three dimensions—Social Identity, Homophily, and Confirmation Bias—and two theoretically related constructs: Online Disinhibition (MOD) and Internet Trolling (GAIT). As shown in Table 5, all IBRS dimensions exhibited positive and statistically significant associations with online disinhibition: Social Identity (r = 0.350, *p* < 0.01), Homophily (r = 0.408, *p* < 0.01) and Confirmation Bias (r = 0.328, *p* < 0.01). Furthermore, modest yet significant correlations were observed between the IBRS dimensions and internet trolling, suggesting that greater identification with virtual communities is related to greater participation in online antisocial behaviours.

### 3.5. Reliability

Both versions of the Identity Bubble Reinforcement Scale (IBRS-9 and IBRS-6) showed adequate internal consistency overall (Table 6). In the IBRS-9, the ω coefficients were high for Social Identity (0.858) and Homophily (0.815) and moderate for Confirmation Bias (0.749). In the abbreviated IBRS-6 version, the first two factors maintained good reliability (ω = 0.848 and 0.818, respectively), whereas Confirmation Bias presented low reliability (ω = 0.600).

## 4. Discussion

The objective of this study was to evaluate the psychometric properties of both the 9-item (IBRS-9) and 6-item (IBRS-6) versions of the Identity Bubble Reinforcement Scale in a large sample of Chilean adolescent students. The results confirmed that both the extended version (IBRS-9) and the short version (IBRS-6) exhibit a stable three-factor structure comprising Social Identity, Homophily, and Confirmation Bias. This configuration is consistent with the original model proposed by Kaakinen et al. [7], reaffirming the scale’s conceptual robustness and its applicability in diverse sociocultural contexts.

Confirmatory factor analyses demonstrated satisfactory fit indices in both versions, whilst the multigroup invariance tests provided evidence of scalar invariance across key demographic groups (sex, social media use, and Internet use). These results indicate that the instrument maintains measurement equivalence across groups, supporting valid comparisons among adolescents with differing digital experiences [73]. Furthermore, the internal consistency indices were satisfactory, evidencing robust psychometric performance for each of the three factors and the overall construct.

In terms of convergent validity, the three dimensions of the IBRS correlated positively and significantly with online disinhibition as measured by the MOD. This result confirms that social identification, homophily, and confirmation bias—all characteristic of identity bubbles—are associated with a greater tendency to act with fewer restrictions in virtual contexts [44,45,73,74], reinforcing the validity of the IBRS in the adolescent population. Furthermore, modest but significant correlations were found between the IBRS dimensions and internet trolling, as measured by the GAIT scale [48]. This finding suggests that a stronger presence of identity bubble dynamics may be associated with an increased propensity to engage in antisocial behaviours in online contexts.

Moreover, item 7 (“In social media, I belong to a community or communities that I can commit to”) showed a lower level of agreement compared to other items, whereas item 9 (“In social media, I can keep myself well informed”) received relatively higher endorsement. These differences may reflect age-related variations in item interpretation, where commitment to virtual communities might be less salient for adolescents, while staying informed through social media represents a more frequent and valued activity at this developmental stage. This pattern underscores the importance of considering developmental and contextual factors when interpreting adolescents’ responses.

Beyond statistical confirmation, these results allow for an integrated understanding of how these three factors influence people’s tendency to become involved in social media identity bubbles [40]. In this sense, it is clear that this behaviour does not arise solely from the use of digital platforms, but is closely linked to psychological and social processes that strengthen a sense of belonging, affinity with others, and the validation of one’s own ideas, thereby creating homogeneous and closed spaces for social interaction [48].

Likewise, this study comparatively analysed the two existing versions of the instrument originally proposed by Kaakinen et al. [7]: the extended nine-item version (IBRS-9) and the abbreviated six-item version (IBRS-6). The results indicated that both versions exhibit an adequate three-factor structure comprising Social Identity, Homophily, and Confirmation Bias, with excellent fit indices and adequate reliability. However, the second-order model showed a slightly more parsimonious fit, indicating the presence of a general identity bubble reinforcement factor that integrates the three specific dimensions. Consistent with the original authors’ findings, the IBRS-9 retains a higher level of theoretical coverage of the construct and a more precise representation of the social identification and homogenisation processes specific to adolescence. Therefore, its use is recommended in research seeking a comprehensive approach to the phenomenon, whereas the IBRS-6 is suitable for brief or large-scale studies where it is necessary to optimise application time without compromising structural validity.

From a broader perspective, the empirical evidence gathered provides a robust theoretical and methodological foundation for further study of Identity Bubble Reinforcement in adolescents [62]. This developmental stage is particularly relevant, as it is characterised by intensive processes of identity construction and the search for belonging, which makes young people especially susceptible to dynamics of homophily, social identification, and informational biases on social media [51]. Therefore, understanding how these identity bubbles operate in this age group contributes to the design of interventions that promote more critical and diverse digital interactions [63].

Regarding the factorial invariance analysis of the IBRS-9, the results show that the instrument’s factorial structure remains stable across all analysed groups (sex, social media use, and internet use). This supports the instrument’s validity for making comparisons between groups. These findings are consistent with previous studies that have reported the invariance of the IBRS-9 instrument [7,62,73].

The reliability evidence for the scale indicates that its items and factors exhibit adequate internal consistency, which supports its use in contexts with an adolescent population [40]. These findings confirm the psychometric stability of the instrument and its relevance for future research aimed at analysing Identity Bubble Reinforcement in adolescents, especially in comparative studies [50].

The present study provides robust empirical evidence regarding the factorial structure, invariance, criterion validity, and reliability of the Identity Bubble Reinforcement Scale in an adolescent population, which reinforces its psychometric soundness. The results obtained allow for a deeper analysis of the relationships between the constructs of social identity, homophily, and confirmation bias in different contexts, favouring its application in future research [7].

### Limitations and Future Research

Despite this study’s contributions, several limitations must be acknowledged. The cross-sectional design limits the causal interpretation of the relationships between identity reinforcement and social media use. Furthermore, the absence of a test–retest analysis prevents the assessment of temporal stability. The inclusion of a limited set of external criteria also limits the exploration of predictive validity. Future research should employ longitudinal designs, incorporate test–retest procedures, and include additional psychosocial and contextual variables, such as digital literacy, parental mediation, and school climate, to refine the explanatory models [74]. The employment of structural equation models could further clarify the direct and indirect effects underlying the emergence and persistence of identity bubbles during adolescence.

In this regard, it is suggested that future research incorporate longitudinal designs that allow for observing how identity bubbles and social media interaction patterns evolve during different moments of adolescent development. This would enable a deeper understanding of the trajectories of influence and the potential long-term effects on the configuration of social identities, as well as exposure to dynamics of homophily and confirmation bias in digital environments.

Another limitation of the abbreviated IBRS-6 version is that the Confirmation Bias factor showed low internal reliability, which reflects the common trade-off between brevity and internal consistency in reduced scales. Replicating these results is recommended to confirm the instrument’s structure and psychometric properties.

Finally, future research could develop broader models that integrate new variables related to identity bubble dynamics, which would allow for a deeper and more detailed analysis of the factors that favour their formation during adolescence [58]. The incorporation of additional variables, such as the level of digital literacy, digital authoritarianism, the quality of the school environment, and family practices related to social media use, would contribute to a more precise understanding of the underlying determinants of this phenomenon [64,75,76,77]. Thus, moving forward with the analysis of more integrative explanatory models would enable testing complex hypotheses using structural equation modelling (SEM). This strategy would allow the identification of direct and indirect relationships among individuals, families, and contextual variables, providing robust evidence for understanding the mechanisms underlying the formation and maintenance of these bubbles in digital environments.

## 5. Conclusions

To sum up, this study provides robust empirical evidence supporting the psychometric soundness of both versions of the Identity Bubble Reinforcement Scale (IBRS-9 and IBRS-6) in adolescents. Both instruments exhibit a consistent three-factor structure and scalar invariance across key groups, confirming their reliability and validity for inter-group comparisons. The longer version is recommended for comprehensive theoretical analyses, whilst the shorter version serves as an efficient tool for large-scale or time-limited assessments.

Furthermore, the invariance analysis showed that the scale’s factorial structure remains stable, reaching the level of scalar invariance. This finding supports the comparability of the measurements at different times, as both the factor loadings and intercepts remained invariant.

In terms of reliability, the internal consistency coefficients obtained were acceptable across all measurements, which supports the instrument’s suitability for use in studies in diverse contexts.

Together, the results support the adequate psychometric quality of the IBRS-9 and IBRS-6, confirming its relevance for research on identity bubbles in social media. Based on its theoretical foundation and empirical validity, this scale is a useful tool for monitoring this phenomenon in adolescent populations. It can help guide future interventions aimed at understanding and addressing the effects of social identification, homophily, and information bias that shape these bubbles.

Finally, the findings of this study reinforce the usefulness of the IBRS-9 as a valid and reliable tool for assessing identity bubbles in adolescents, contributing to the advancement of knowledge in this field and offering a solid foundation for future research.

## Figures and Tables

**Table 1 children-12-01545-t001:** Main Characteristics of the Sample.

Variables	Categories	n (%)
Sex	Male	50.8 (%)
	Female	47.8 (%)
	Other	1.4 (%)
Internet Use	Between 1 and 4 h	45.3 (%)
	More than 5 h	54.7 (%)
Social Media Use	Between 1 and 4 h	59.8 (%)
	More than 5 h	40.2 (%)
Ethnicity	Indigenous (aymara/mapuche)	20.6 (%)
	Non-Indigenous	79.4 (%)

**Table 2 children-12-01545-t002:** Descriptive Statistics.

Items	M	Md	Sd	IQR	g_1_	g_2_
It1	2.39	2.00	1.30	2.00	0.50	−0.98
It2	2.51	2.00	1.31	3.00	0.33	−1.10
It3	3.18	3.0	1.24	2.00	−0.38	−0.86
It4	3.51	4.00	1.16	1.00	−0.71	−0.28
It5	2.42	2.00	1.05	1.00	0.30	−0.56
It6	2.38	2.00	1.04	1.50	0.22	−0.70
It7	2.35	2.00	1.21	2.00	0.47	−0.86
It8	3.28	4.00	1.20	1.00	−0.51	−0.62
It9	3.85	4.00	1.04	2.00	−1.06	0.84

Note. M = mean; Md = median; Sd = standard deviation; IQR = interquartile range; g_1_ = skewness; g_2_ = kurtosis; K–S = Kolmogorov–Smirnov test. *p* < 0.001.

**Table 3 children-12-01545-t003:** Comparison of Competing Measurement Models for the IBRS-9 and IBRS-6.

Scale	Model	χ^2^	df	CFI	TLI	RMSEA	SRMR	ΔCFI vs. 3F	ΔRMSEA vs. 3F	DIFFTEST χ^2^	DIFFTEST df	DIFFTEST p	Notes
IBRS-9	3 correlated factors	429.746 *	24	0.978	0.967	0.061	0.044	—	—	—	—	—	Baseline model—Accepted
	Second-order (G over 3)	448.947 *	25	0.977	0.967	0.061	0.045	−0.001	0.000	34.960 *	1	<0.001	Statistically different fit, but negligible ΔCFI—Accepted
IBRS-6	3 correlated factors	31.521 *	6	0.998	0.994	0.031	0.013	—	—	—	—	—	Baseline model—Accepted
	Second-order (G over 3)	52.980 *	7	0.996	0.991	0.038	0.018	−0.002	0.007	15.950 *	1	0.0001	Statistically different fit, but negligible ΔCFI—Accepted

Note: *χ*^2^ = Chi-square goodness of fit test (asterisk indicates *p* < 0.01); df = degrees of freedom; CFI = Comparative Fit Index; TLI = Tucker–Lewis Index; RMSEA = Root Mean Square Error of Approximation; SRMR = standardised Root Mean Square Residual; ΔCFI/ΔRMSEA = difference relative to the 3-factor model; DIFFTEST = robust chi-square difference test for WLSMV (nested models); Models showing excellent fit and very small differences in CFI—equal to or less than two thousandths—were considered accepted. Models showing inadequate or poor fit were rejected.

**Table 4 children-12-01545-t004:** Evaluation of measurement invariance across groups.

Scale	Variable/Model	WLSMV-χ^2^ (df)	RMSEA	CFI	TLI	SRMR	ΔRMSEA	ΔCFI	ΔTLI	DECISIÓN
Sex										
IBRS-9	M0	601.834 (48)	0.073	0.981	0.971	0.044	—	—	—	Accepted
	M1	501.380 (57)	0.060	0.984	0.980	0.046	−0.013	+0.003	+0.009	Accepted
	M2	617.089 (60)	0.065	0.980	0.977	0.045	+0.005	−0.004	−0.003	Accepted
	M3	570.407 (69)	0.058	0.982	0.982	0.045	−0.007	+0.002	+0.005	Accepted
IBRS-6	M0	34.879 (12)	0.030	0.999	0.997	0.013	—	—	—	Accepted
	M1	62.873 (18)	0.034	0.998	0.996	0.016	+0.004	−0.001	−0.001	Accepted
	M2	70.808 (18)	0.037	0.997	0.995	0.016	+0.003	−0.001	−0.001	Accepted
	M3	74.839 (24)	0.031	0.997	0.997	0.015	−0.006	0.000	+0.002	Accepted
Social media use										
IBRS-9	M0	589.749 (48)	0.071	0.981	0.972	0.044	—	—	—	Accepted
	M1	451.242 (57)	0.056	0.986	0.983	0.044	−0.015	+0.005	+0.011	Accepted
	M2	571.311 (60)	0.062	0.982	0.979	0.044	+0.006	−0.004	−0.004	Accepted
	M3	523.340 (69)	0.054	0.984	0.984	0.044	−0.008	+0.002	+0.005	Accepted
IBRS-6	M0	40.066 (12)	0.032	0.999	0.996	0.013	—	—	—	Accepted
	M1	34.801 (18)	0.021	0.999	0.999	0.015	−0.011	0.000	+0.003	Accepted
	M2	44.340 (18)	0.026	0.999	0.998	0.014	+0.005	0.000	−0.001	Accepted
	M3	46.880 (24)	0.021	0.999	0.998	0.015	−0.005	0.000	0.000	Accepted
Internet use										
IBRS-9	M0	597.244 (48)	0.073	0.981	0.971	0.045	—	—	—	Accepted
	M1	467.886 (57)	0.058	0.986	0.982	0.046	−0.015	+0.005	+0.011	Accepted
	M2	585.258 (60)	0.064	0.982	0.978	0.045	+0.006	−0.004	−0.004	Accepted
	M3	533.530 (69)	0.056	0.984	0.983	0.045	−0.008	+0.002	+0.005	Accepted
IBRS-6	M0	36.911 (12)	0.031	0.999	0.997	0.013	—	—	—	Accepted
	M1	38.337 (18)	0.023	0.999	0.998	0.014	−0.008	0.000	+0.001	Accepted
	M2	40.342 (18)	0.024	0.999	0.998	0.014	+0.001	0.000	0.000	Accepted
	M3	43.703 (23)	0.019	0.999	0.999	0.014	−0.005	0.000	+0.001	Accepted
Age										
IBRS-9	M0	608.681 (48)	0.072	0.981	0.971	0.044	—	—	—	Accepted
	M1	465.128 (57)	0.056	0.986	0.983	0.044	−0.016	+0.005	+0.012	Accepted
	M2	591.483 (60)	0.063	0.982	0.978	0.044	+0.007	−0.004	−0.005	Accepted
	M3	542.425 (69)	0.055	0.984	0.983	0.044	−0.008	+0.002	+0.005	Accepted
IBRS-6	M0	38.211 (12)	0.031	0.999	0.997	0.013	—	—	—	Accepted
	M1	38.346 (18)	0.022	0.999	0.998	0.015	−0.009	0.000	+0.001	Accepted
	M2	37.827 (18)	0.022	0.999	0.998	0.013	0.000	0.000	0.000	Accepted
	M3	49.245 (24)	0.022	0.999	0.998	0.014	0.000	0.000	0.000	Accepted

Note: WLSMV-χ^2^ = Chi-square statistic estimated with the robust weighted least squares mean and variance adjusted estimator; df = degrees of freedom; RMSEA = root mean square error of approximation; CFI = comparative fit index; TLI = Tucker–Lewis index; SRMR = standardised root mean square residual; ΔRMSEA, ΔCFI, and ΔTLI = differences between successive nested models (M1–M0, M2–M1, M3–M2). Model sequence: M0 = configural invariance; M1 = metric invariance; M2 = scalar invariance; M3 = strict invariance. Invariance decisions were based on the criteria proposed by Chen [69]: ΔCFI ≤ 0.010, ΔTLI ≤ 0.010, and ΔRMSEA ≤ 0.015, indicating acceptable invariance between groups. IBRS-9 = Identity Bubble Reinforcement Scale (9-item version); IBRS-6 = Identity Bubble Reinforcement Scale (6-item version). Positive and negative signs indicate the direction of change in the fit indices.

**Table 5 children-12-01545-t005:** Correlations Between Online Disinhibition and Identity Bubble Dimensions.

Factors	Online Disinhibition	Internet Trolling	Social Identity	Homophily	Confirmation Bias
Online Disinhibition	1.00				
Internet Trolling	0.395 **	1.00			
Social Identity	0.350 **	0.194 **	1.00		
Homophily	0.408 **	0.054 **	0.371 **	1.00	
Confirmation Bias	0.328 **	0.139 **	0.432 **	0.350 **	1.00

Note. ** indicates *p* < 0.01 (two-tailed). Online Disinhibition = Measure of Online Disinhibition (MOD); Internet Trolling = Global Assessment of Internet Trolling (GAIT); Social Identity, Homophily, and Confirmation Bias are the three factors of the Identity Bubble Reinforcement Scale (IBRS).

**Table 6 children-12-01545-t006:** Internal Consistency and Reliability Estimates for the IBRS-9 and IBRS-6 Factors.

**IBRS-9**	**Factors**	**McDonald’s ω**	**IC 95% ω**	**Cronbach’s α**	**IC 95% α**
	Social Identity	0.858	[0.851–0.866]	0.853	[0.845–0.861]
	Homophily	0.815	[0.802–0.827]	0.810	[0.799–0.819]
	Confirmation Bias	0.749	[0.730–0.767]	0.720	[0.694–0.745]
**IBRS-6**	**Factors**				
	Social Identity	0.848	[0.836–0.861]	0.849	[0.791–0.811]
	Homophily	0.818	[0.802–0.831]	0.818	[0.807–0.829]
	Confirmation Bias	0.600	[0.570–0.627]	0.600	[0.575–0.624]

Note. Reliability estimates are presented for the three latent factors of the Identity Bubble Reinforcement Scale (IBRS-9 and IBRS-6 versions). McDonald’s Omega coefficients (ω) and Cronbach’s alpha (α) are reported with their respective 95% confidence intervals (CI).

## Data Availability

The dataset for the study is available from the corresponding author upon reasonable request due to ethical restrictions.

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
