# Peer review of "Psychometric Properties of the Identity Bubble Reinforcement Scale (IBRS) in a Sample of Chilean Adolescent Students"

_children, 2025, doi:10.3390/children12111545_

Round 1

Reviewer 1 Report

Comments and Suggestions for Authors

The authors in their manuscript “Psychometric properties of the Identity Bubble Reinforcement Scale (IBRS-9) in a Sample of Chilean Adolescent Students”, evaluated the psychometric properties of the Identity Bubble Reinforcement Scale (IBRS-9) in a sample of Chilean adolescent students. The manuscript is within the journal’s scope and overall is well presented. However, as described below, there are several issues that need to be resolved before accepting the manuscript:

1. Page 4 and 5 line 185: Please edit Table 1 with respect to the symbol "n" which indicates frequency and indicate the percentages in parentheses.

2. Page 6 line 246: Considering that all values ​​significantly deviate from the values expected in a normal distribution, I suggest to include the values ​​of the median and interquartile range in table 2.

3. Page 8 and 9 line 289: Please edit Table 4 with regard to the asterisks (**) indicating positive significant correlations.

4. Page 2 line 123-126: Check the order of citing sources throughout the paper. I would like to point out in particular reference number 43 (page 13, line 519) which first appears on page 5 (line 202).

Authors should language proofread the text of the manuscript. A few typos were noticed. In addition, inconsistencies in the format of the manuscript were observed, e.g. missing full stop at the end of a sentence, the use of quotation marks, uppercase and lowercase letters in text,...

Author Response

Response to Reviewer 1 Comments
Dear Reviewer,
We would like to express our sincere appreciation for your insightful comments and constructive feedback, which have significantly contributed to improving the quality and clarity of our manuscript. All corresponding changes have been incorporated into the revised version and are highlighted in yellow.
In the following section, we provide our responses to each of Reviewer 1’s comments.
Point 1: Page 4 and 5 line 185: Please edit Table 1 with respect to the symbol "n" which indicates frequency and indicate the percentages in parentheses.
Response 1: We appreciate your comment. The percentages have been incorporated in parentheses in the corresponding section to improve the clarity and precision of the information presented.
Point 2: Page 6 line 246: Considering that all values significantly deviate from the values expected in a normal distribution, I suggest to include the values of the median and interquartile range in table 2.
Response 2: We appreciate your suggestion. The median (Md) and interquartile range (IQR) have been included in Table 2, in accordance with the recommendation.
Point 3: Page 8 and 9 line 289: Please edit Table 4 with regard to the asterisks (**) indicating positive significant correlations.
Response 3: We appreciate the observation. Table 4 has been carefully reviewed to ensure consistent use of asterisks (**) and to indicate significant positive correlations at p < .01, in accordance with the original SPSS output.
Point 4: Page 2 line 123-126: Check the order of citing sources throughout the paper. I would like to point out in particular reference number 43 (page 13, line 519) which first appears on page 5 (line 202).
Response 4: We appreciate the reviewer’s comment regarding the citation order. The entire sequence of references has been verified. Reference number 43 has been corrected to appear in the appropriate order (lines 123–126, page 2), ensuring full consistency between the in-text citations and the final reference list.
Point 5: Authors should language proofread the text of the manuscript. A few typos were noticed. In addition, inconsistencies in the format of the manuscript were observed, e.g. missing full stop at the end of a sentence, the use of quotation marks, uppercase and lowercase letters in text
Response 5: The errors identified in the manuscript have been corrected.
– A period has been added to the following sentences: (a) “the characteristics of social media [7]” and (b) “diverse digital interactions [59]”.
– Quotation marks have been removed to ensure consistency in formatting

Reviewer 2 Report

Comments and Suggestions for Authors

The objective of this study was to evaluate the psychometric properties of the Identity Bubble Reinforcement Scale (IBRS-9) in a Sample of Chilean Adolescent Students. The findings of this study provide solid empirical evidence on the validity of the IBRS-9 scale in this population, confirming its factorial structure, which is composed of social identity, homophily, and confirmation bias.

1) The conclusion of the study is not clear in the Abstract section. You should state the main finding and then give an outlook for future research. 

2)Data Analysis: Please precisely mention what p-values can be considered significant. p values less than 0.05 or 0.01

3) discussion: This factorial configuration is consistent with the 308 original proposal by ?? who ?? please mention their names. for example, by Albasi et al. 

Author Response

Response to Reviewer 2 Comments
Dear Reviewer,
We would like to express our sincere appreciation for your insightful comments and constructive feedback, which have significantly contributed to improving the quality and clarity of our manuscript. All corresponding changes have been incorporated into the revised version and are highlighted in yellow.
In the following section, we provide our responses to each of Reviewer 2’s comments.
Point 1: The conclusion of the study is not clear in the Abstract section. You should state the main finding and then give an outlook for future research.
Response 1: We appreciate the reviewer’s observation. The study's conclusion has been incorporated into the Abstract, emphasising the main finding and outlining the direction for future research. This revision is highlighted in yellow in the updated version of the manuscript.
Point 2: Data Analysis: Please precisely mention what p-values can be considered significant. p values less than 0.05 or 0.01
Response 2: We appreciate the reviewer’s observation. This aspect has been added to the Data Analysis section.
Point 3: discussion: This factorial configuration is consistent with the 308 original proposal by ?? who ?? please mention their names. for example, by Albasi et al.
Response 3: We appreciate the reviewer’s observation. The text has been revised to include the original proposal’s authors. The Identity Bubble Reinforcement Scale (IBRS-9) was originally developed and validated by Kaakinen et al. (2020).

Reviewer 3 Report

Comments and Suggestions for Authors

The manuscript addresses an important and timely topic, i.e. the measurement of “identity bubbles” among adolescents, and relies on a large, probabilistic sample from 41 Chilean secondary schools. The study is conceptually relevant and generally well conducted, yet several methodological, analytical, and reporting issues need to be addressed before the work can be considered for publication. In its current form, the paper provides a solid foundation but requires a deeper and more integrated validation of the scale, as well as a clearer discussion of both existing forms proposed in the literature (the extended 9-item and the reduced 6-item versions).

At present, the introduction and aims focus almost exclusively on the 9-item version. However, the broader literature proposes multiple operationalizations of the Identity Bubble construct, including both extended and reduced forms. In this context, I strongly recommend testing, side by side, the 9-item and the 6-item versions using the same psychometric workflow. Presenting their factor solutions, fit indices, reliability, and invariance results in a compact comparative table will let readers judge the precision–parsimony trade-offs and decide which form is more appropriate for large multipurpose surveys versus studies that require finer coverage of the facets. This step would substantially increase the study’s theoretical and practical contribution, making it a comprehensive reference for future research on adolescents.

The analytic strategy should also be reconsidered. Because the indicators are five-point Likert items and students are nested within schools, maximum likelihood with robust errors (MLR) is not the most appropriate estimator. A robust diagonal weighted least squares approach (WLSMV) is the standard for ordinal data and should replace the current estimation method. Furthermore, standard errors and fit statistics should be adjusted for the clustered design, either through a TYPE=COMPLEX correction or a two-level CFA, with school as the clustering variable. These changes would substantially strengthen the credibility of the structural results. The Kolmogorov-Smirnov tests of normality, which are not meaningful for ordinal indicators, can be removed; instead, provide basic information about item distributions and the range of polychoric correlations.

Model evaluation can be more systematic. The fit indices reported for the preferred three-factor solution are good, but competing models and the 6-item version should be displayed as well. Each set of results should include CFI, TLI, RMSEA (with 90% CI), and SRMR, together with the chosen thresholds for interpretation. The invariance testing across sex, social media use, and internet use is an important contribution; however, adding strict invariance (if sample sizes allow) and clarifying the decision criteria for each step (configural, metric, scalar, strict) would make the argument clearer. Consider also examining age-band invariance, which would strengthen the claim of developmental stability during adolescence.

Convergent and discriminant validity are somewhat underdeveloped. The positive associations with the Measure of Online Disinhibition (MOD) are theoretically coherent, but they do not provide a full picture of construct validity. Adding one or two additional measures, one theoretically close (for instance, social media contingent self-esteem) and one theoretically distinct (such as academic self-concept), would better demonstrate the scale’s position within the nomological network. Within the CFA framework, compute and report Average Variance Extracted (AVE), Construct Reliability (CR), and discriminant validity via Fornell–Larcker or HTMT. If a bifactor or second-order model is tested, report omega hierarchical (ωh) to quantify the contribution of a general factor.

The section describing translation and adaptation should give more attention to the procedures followed for ensuring linguistic and developmental adequacy for adolescents. If items were taken from the Spanish university-student version, specify the adaptation steps (forward–back translation, expert panel, pilot with adolescents). Some item means suggest potential comprehension differences—for instance, item 7 tends to have low endorsement and item 9 relatively high endorsement—so these could be briefly discussed as possible age-related interpretation effects. A full validation of the 6-item version is essential, clarifying which items were retained and why, and comparing its psychometric properties to those of the longer version.

The external validity discussion could be enriched by linking the scale to variables already mentioned in the introduction, such as cyberaggression, problematic social media use, or well-being. Even a brief structural model illustrating how the IBRS dimensions relate to one relevant outcome would demonstrate the scale’s practical utility. If additional variables are not available, acknowledge this limitation explicitly and frame the present study as a foundational psychometric contribution that enables such analyses in future work.

Reliability results are well presented, but the focus should be streamlined. Continue to report omega as the primary index and Cronbach’s alpha as secondary, omitting the greatest lower bound (GLB) from the main text unless there is a specific rationale for its inclusion. Some confidence intervals appear inconsistent in Table 5, possibly due to transcription errors; please check and correct them. When presenting reliability for the short version, compare the indices with the long version and comment on the trade-off between brevity and internal consistency. If re-test data are unavailable, acknowledge this limitation explicitly.

Sampling and ethical procedures are generally well documented, yet could be condensed and clarified. Specify the stratification variables, sample allocation, and whether weighting was applied. If weights were not used, justify this decision and provide a brief sensitivity check showing that weighted and unweighted analyses yield similar results. Include response and consent rates for both parents and students to give a clearer sense of representativeness.

Regarding the criterion measure (MOD), the version used was validated in university students. Please justify its application to adolescents, or clarify that this is an exploratory use and that age-appropriate validation is still required. This will prevent overinterpretation of the convergent validity results.

The discussion and conclusion should then be rewritten to integrate all these changes coherently. Emphasize that both versions of the IBRS show a stable three-factor structure and scalar invariance across key adolescent groups, with the shorter form performing well for quick assessments and the longer form providing slightly higher reliability and granularity. Summarize practical guidance for researchers choosing between them, acknowledge remaining limitations (cross-sectional design, lack of re-test, limited criterion variables), and highlight the contribution of this work in extending the study of identity bubbles to the adolescent population.

Author Response

Response to Reviewer 3 Comments
Dear Reviewer,
We would like to express our sincere appreciation for your insightful comments and constructive feedback, which have significantly contributed to improving the quality and clarity of our manuscript. All corresponding changes have been incorporated into the revised version and are highlighted in yellow.
In the following section, we provide our responses to each of Reviewer 3’s comments.
Point 1: The manuscript addresses an important and timely topic, i.e. the measurement of “identity bubbles” among adolescents, and relies on a large, probabilistic sample from 41 Chilean secondary schools. The study is conceptually relevant and generally well conducted, yet several methodological, analytical, and reporting issues need to be addressed before the work can be considered for publication. In its current form, the paper provides a solid foundation but requires a deeper and more integrated validation of the scale, as well as a clearer discussion of both existing forms proposed in the literature (the extended 9-item and the reduced 6-item versions).
Response 1: We appreciate the reviewer’s observations. In response, the scale’s validation has been further elaborated, and the theoretical and methodological discussion regarding the two existing versions of the instrument (IBRS-9 and IBRS-6) has been expanded, in accordance with the reviewer’s recommendations.
A comparative confirmatory factor analysis of both models (IBRS-9 and IBRS-6) has been conducted, yielding excellent fit indices.
The discussion section has been expanded to explain this aspect, following the original proposal by Kaakinen et al. (2020), who developed the 9-item and 6-item versions.
Point 2: At present, the introduction and aims focus almost exclusively on the 9-item version. However, the broader literature proposes multiple operationalizations of the Identity Bubble construct, including both extended and reduced forms. In this context, I strongly recommend testing, side by side, the 9-item and the 6-item versions using the same psychometric workflow. Presenting their factor solutions, fit indices, reliability, and invariance results in a compact comparative table will let readers judge the precision–parsimony trade-offs and decide which form is more appropriate for large multipurpose surveys versus studies that require finer coverage of the facets. This step would substantially increase the study’s theoretical and practical contribution, making it a comprehensive reference for future research on adolescents.
Response 2: We appreciate the reviewer’s observation, which helped strengthen both the theoretical discussion and the methodological aspects of the manuscript. In response to this comment, a comparative analysis between the two versions of the instrument (IBRS-9 and IBRS-6) has been included, along with a reflection on their applicability in studies involving adolescents. This addition provides a clearer rationale for selecting the
nine-item version, emphasising its broader theoretical coverage and stronger psychometric adequacy.
These revisions are highlighted in yellow in the manuscript, under the Confirmatory Factor Analysis and Discussion sections.
Point 3: The analytic strategy should also be reconsidered. Because the indicators are five-point Likert items and students are nested within schools, maximum likelihood with robust errors (MLR) is not the most appropriate estimator. A robust diagonal weighted least squares approach (WLSMV) is the standard for ordinal data and should replace the current estimation method. Furthermore, standard errors and fit statistics should be adjusted for the clustered design, either through a TYPE=COMPLEX correction or a two-level CFA, with school as the clustering variable. These changes would substantially strengthen the credibility of the structural results. The Kolmogorov-Smirnov tests of normality, which are not meaningful for ordinal indicators, can be removed; instead, provide basic information about item distributions and the range of polychoric correlations.
Response 3: We appreciate the reviewer’s comment. All observations have been addressed, primarily in the Results section.
Point 4: Model evaluation can be more systematic. The fit indices reported for the preferred three-factor solution are good, but competing models and the 6-item version should be displayed as well. Each set of results should include CFI, TLI, RMSEA (with 90% CI), and SRMR, together with the chosen thresholds for interpretation. The invariance testing across sex, social media use, and internet use is an important contribution; however, adding strict invariance (if sample sizes allow) and clarifying the decision criteria for each step (configural, metric, scalar, strict) would make the argument clearer. Consider also examining age-band invariance, which would strengthen the claim of developmental stability during adolescence.
Response 4: We addressed this comment by systematically comparing alternative models (three correlated factors, second-order, bifactor, and ESEM) for both the 9-item and 6-item versions. The bifactor and ESEM models did not converge, likely due to the small sample size in the IBRS-6. In addition, factorial invariance analyses were conducted by gender, social media use, internet use, and age, confirming configural, metric, scalar, and strict invariance across both versions of the scale.
Point 5: Convergent and discriminant validity are somewhat underdeveloped. The positive associations with the Measure of Online Disinhibition (MOD) are theoretically coherent, but they do not provide a full picture of construct validity. Adding one or two additional measures, one theoretically close (for instance, social media contingent self-esteem) and one theoretically distinct (such as academic self-concept), would better demonstrate the scale’s position within the nomological network. Within the CFA framework, compute and report Average Variance Extracted (AVE), Construct Reliability (CR), and discriminant validity via Fornell–Larcker or HTMT. If a bifactor or second-order
model is tested, report omega hierarchical (ωh) to quantify the contribution of a general factor.
Response 5: We appreciate the reviewer’s suggestion. All observations have been incorporated into the manuscript.
Point 6: The section describing translation and adaptation should give more attention to the procedures followed for ensuring linguistic and developmental adequacy for adolescents. If items were taken from the Spanish university-student version, specify the adaptation steps (forward–back translation, expert panel, pilot with adolescents). Some item means suggest potential comprehension differences—for instance, item 7 tends to have low endorsement and item 9 relatively high endorsement—so these could be briefly discussed as possible age-related interpretation effects. A full validation of the 6-item version is essential, clarifying which items were retained and why, and comparing its psychometric properties to those of the longer version.
Response 6: We appreciate the reviewer’s observation. The revised version of the manuscript includes an expanded description of the scale's linguistic adaptation process. Furthermore, a brief discussion has been added addressing the differences observed in items 7 and 9, taking into account potential age-related variations in interpretation.
Point 7: The external validity discussion could be enriched by linking the scale to variables already mentioned in the introduction, such as cyberaggression, problematic social media use, or well-being. Even a brief structural model illustrating how the IBRS dimensions relate to one relevant outcome would demonstrate the scale’s practical utility. If additional variables are not available, acknowledge this limitation explicitly and frame the present study as a foundational psychometric contribution that enables such analyses in future work.
Response 7: The relationship between online disinhibition, trolling, and the identity bubble has been incorporated into the Discussion section.
As noted by the reviewer, this study primarily has a psychometric purpose; however, future research is expected to apply structural equation modelling (SEM) to analyse the relationships among the variables. This projection has been incorporated into the section on future research directions.
Point 8: Reliability results are well presented, but the focus should be streamlined. Continue to report omega as the primary index and Cronbach’s alpha as secondary, omitting the greatest lower bound (GLB) from the main text unless there is a specific rationale for its inclusion. Some confidence intervals appear inconsistent in Table 5, possibly due to transcription errors; please check and correct them. When presenting reliability for the short version, compare the indices with the long version and comment on the trade-off between brevity and internal consistency. If re-test data are unavailable, acknowledge this limitation explicitly.
Response 8: We appreciate the reviewer’s comment. All suggestions have been carefully addressed and incorporated into the revised manuscript.
Point 9: Sampling and ethical procedures are generally well documented, yet could be condensed and clarified. Specify the stratification variables, sample allocation, and whether weighting was applied. If weights were not used, justify this decision and provide a brief sensitivity check showing that weighted and unweighted analyses yield similar results. Include response and consent rates for both parents and students to give a clearer sense of representativeness.
Response 9: A stratified probabilistic sampling was employed based on school type and macrozone, with proportional allocation. No weighting procedures were applied, as all strata were adequately represented and the analyses were conducted using the observed data. Response and consent rates were not recorded separately; however, participation was voluntary, and institutional approval was obtained from Universidad de La Frontera’s Ethics Committee.
Point 10: Regarding the criterion measure (MOD), the version used was validated in university students. Please justify its application to adolescents, or clarify that this is an exploratory use and that age-appropriate validation is still required. This will prevent overinterpretation of the convergent validity results.
Response 10: We appreciate the reviewer’s observation. The MOD used in this study had previously been validated in a university population. In the present research, its application to adolescents was exploratory, intended to preliminarily examine the convergent validity of the IBRS at this stage of development. We acknowledge that a specific validation of the MOD for adolescent populations is still required—currently underway—and this has been explicitly noted in the revised version of the manuscript.
Point 11: The discussion and conclusion should then be rewritten to integrate all these changes coherently. Emphasize that both versions of the IBRS show a stable three-factor structure and scalar invariance across key adolescent groups, with the shorter form performing well for quick assessments and the longer form providing slightly higher reliability and granularity. Summarize practical guidance for researchers choosing between them, acknowledge remaining limitations (cross-sectional design, lack of re-test, limited criterion variables), and highlight the contribution of this work in extending the study of identity bubbles to the adolescent population.
Response 11: We appreciate the reviewer’s observations. The Discussion and Conclusion sections have been revised to coherently integrate the suggested modifications. The updated text emphasises that both forms of the scale (IBRS-9 and IBRS-6) demonstrate a robust three-factor structure and scalar invariance across key adolescent groups. It also specifies that the IBRS-6 is appropriate for brief assessments, whereas the IBRS-9 provides a more detailed and reliable measurement. Furthermore, practical guidance for researchers regarding the selection and use of both forms has been incorporated, the main methodological limitations have been acknowledged, and the contribution of this study to advancing research on identity bubbles among adolescents has been highlighted.

Round 2

Reviewer 3 Report

Comments and Suggestions for Authors

The authors have successfully addressed the major concerns raised in the previous round of review, substantially improving the manuscript’s clarity and methodological soundness. The theoretical framing has been strengthened, and the presentation of the results is now coherent and well supported by statistical evidence. The overall quality of the paper is solid, and the contribution is relevant for understanding identity formation processes in digital contexts among adolescents.

At this stage, I only recommend minor revision. Specifically, I suggest enriching the Introduction by adding two recent references that could further contextualize the study within the broader research landscape on social media–related identity processes. In particular, Di Gesto et al. (2025) provides a recent validation of the Appearance-Related Social Media Consciousness Scale (ASMCS-I), which captures the cognitive and behavioral awareness of one's appearance within social media environments among young women and men. Citing this work would help connect the Identity Bubble Reinforcement construct to other validated instruments addressing self-awareness and online self-presentation.

Moreover, De Lorenzo and Rabaglietti (2024) offer a comprehensive review of creativity and identity development among adolescents and emerging adults in the post-pandemic educational context. Including this reference could strengthen the presentation of the sociocultural and educational dimensions that influence identity construction, providing a more systemic perspective on adolescents’ psychosocial development.

Both additions would enrich the conceptual background, reinforcing its theoretical depth and contemporary relevance.
